# Fairness in Learning: Classic and Contextual Bandits <sup>*</sup>

**Matthew Joseph**       **Michael Kearns**       **Jamie Morgenstern**       **Aaron Roth**

University of Pennsylvania, Department of Computer and Information Science
`majos, mkearns, jamiemor, aaroth@cis.upenn.edu`

## Abstract

We introduce the study of fairness in multi-armed bandit problems. Our fairness definition demands that, given a pool of applicants, a worse applicant is never favored over a better one, despite a learning algorithm's uncertainty over the true payoffs. In the classic stochastic bandits problem we provide a provably fair algorithm based on "chained" confidence intervals, and prove a cumulative regret bound with a cubic dependence on the number of arms. We further show that any fair algorithm must have such a dependence, providing a strong separation between fair and unfair learning that extends to the general contextual case. In the general contextual case, we prove a tight connection between fairness and the KWIK (Knows What It Knows) learning model: a KWIK algorithm for a class of functions can be transformed into a provably fair contextual bandit algorithm and vice versa. This tight connection allows us to provide a provably fair algorithm for the linear contextual bandit problem with a polynomial dependence on the dimension, and to show (for a different class of functions) a worst-case exponential gap in regret between fair and non-fair learning algorithms.

## 1   Introduction

Automated techniques from statistics and machine learning are increasingly being used to make decisions that have important consequences on people's lives, including hiring [24], lending [10], policing [25], and even criminal sentencing [7]. These high stakes uses of machine learning have led to increasing concern in law and policy circles about the potential for (often opaque) machine learning techniques to be *discriminatory* or *unfair* [13, 6]. At the same time, despite the recognized importance of this problem, very little is known about technical solutions to the problem of "unfairness", or the extent to which "fairness" is in conflict with the goals of learning.

In this paper, we consider the extent to which a natural fairness notion is compatible with learning in a *bandit* setting, which models many of the applications of machine learning mentioned above. In this setting, the learner is a sequential decision maker, which must choose at each time step $t$ which decision to make from a finite set of $k$ "arms". The learner then observes a stochastic *reward* from (only) the arm chosen, and is tasked with maximizing total earned reward (equivalently, minimizing total *regret*) by learning the relationships between arms and rewards over time. This models, for example, the problem of learning the association between loan applicants and repayment rates over time by repeatedly granting loans and observing repayment.

We analyze two variants of the setting: in the *classic* case, the learner's only source of information comes from choices made in previous rounds. In the *contextual case*, before each round the learner additionally observes some potentially informative *context* for each arm (for example representing the content of an individual's loan application), and the expected reward is some unknown function of

---

<sup>*</sup>A full technical version of this paper is available on arXiv [17].

the context. The difficulty in this task stems from the unknown relationships between arms, rewards, and (in the contextual case) contexts: these relationships must be learned.

We introduce fairness into the bandit learning framework by saying that it is *unfair* to preferentially choose one arm over another if the chosen arm has lower expected quality than the unchosen arm. In the loan application example, this means that it is unfair to preferentially choose a less-qualified applicant (in terms of repayment probability) over a more-qualified applicant.

It is worth noting that this definition of fairness (formalized in the preliminaries) is entirely consistent with the optimal policy, which can simply choose at each round to play uniformly at random from the arms maximizing the expected reward. This is because – it seems – this definition of fairness is entirely consistent with the goal of maximizing expected reward. Indeed, the fairness constraint exactly states that the algorithm *cannot* favor low reward arms!

Our main conceptual result is that this intuition is incorrect in the face of unknown reward functions. Although fairness is consistent with *implementing* the optimal policy, it may not be consistent with *learning* the optimal policy. We show that fairness always has a cost, in terms of the achievable learning rate of the algorithm. For some problems, the cost is mild, but for others, the cost is large.

## 1.1  Our Results

We divide our results into two parts. First, in Section 3 we study the classic stochastic multi-armed bandit problem [20, 19]. In this case, there are no contexts, and each arm $i$ has a fixed but unknown average reward $\mu_i$. In Section 3.1 we give a fair algorithm, FAIRBANDITS, and show that it guarantees nontrivial regret after $T = O(k^3)$ rounds. We then show in Section 3.2 that it is not possible to do better – *any* fair learning algorithm can be forced to endure constant per-round regret for $T = \Omega(k^3)$ rounds, thus tightly characterizing the optimal regret attainable by fair algorithms in this setting, and formally separating it from the regret attainable by algorithms absent a fairness constraint.

We then move on to the general contextual bandit setting in Section 4 and prove a broad characterization result, relating fair contextual bandit learning to *KWIK* ("Knows What It Knows") learning [22]. Informally, a KWIK leaarning algorithm receives a series of unlabeled examples and must either predict a label or announce "I Don't Know". The KWIK requirement then stipulates that any predicted must be label close to its true label. The quality of a KWIK learning algorithm is characterized by its "KWIK bound", which provides an upper bound on the maximum number of times the algorithm can be forced to announce "I Don't Know". For any contextual bandit problem (defined by the set of functions $C$ from which the payoff functions may be selected), we show that the optimal learning rate of any fair algorithm is determined by the best KWIK bound for the class $C$. We prove this constructively via a reduction showing how to convert a KWIK learning algorithm into a fair contextual bandit algorithm in Section 4.1, and vice versa in Section 4.2.

This general connection immediately allows us to import known results for KWIK learning [22]. It implies that some fair contextual bandit problems are *easy* and achieve non-trivial regret guarantees in only polynomial many rounds. Conversely, it also implies that some contextual bandit problems which are easy without the fairness constraint become *hard* once we impose the fairness constraint, in that any fair algorithm must suffer constant per-round regret for exponentially many rounds. By way of example, we will show in Section 4.1 that real contexts with linear reward functions are easy, and we will show in Section 4.3 that boolean context vectors and conjunction reward functions are hard.

## 1.2  Other Related Work

Many papers study the problem of fairness in machine learning. One line of work studies algorithms for batch classification which achieve *group fairness* otherwise known as *equality of outcomes*, *statistical parity* – or algorithms that avoid *disparate impact* (see e.g. [11, 23, 18, 15, 16] and [2] for a study of *auditing* existing algorithms for disparate impact). While statistical parity is sometimes a desirable or legally required goal, as observed by Dwork et al. [14] and others, it suffers from a number of drawbacks. First, if different populations indeed have different statistical properties, then it can be at odds with accurate classification. Second, even in cases when statistical parity is attainable with an optimal classifier, it does not prevent discrimination at an individual level. This led  Dwork et al. [14] to encourage the study of *individual fairness*, which we focus on here.

Dwork et al. [14] also proposed and explored a technical definition of individual fairness formalizing the idea that "similar individuals should be treated similarly" by presupposing a task-specific quality metric on individuals and proposing that fair algorithms should satisfy a Lipschitz condition on this metric. Our definition of fairness is similar, in that the expected reward of each arm is a natural metric through which we define fairness. However, where Dwork et al. [14] presupposes the existence of a "fair" metric on individuals – thus encoding much of the relevant challenge, as studied Zemel et al. [27] – our notion of fairness is entirely aligned with the goal of the algorithm designer and is satisfied by the optimal policy. Nevertheless, it affects the space of feasible learning algorithms, because it interferes with *learning* an optimal policy, which depends on the unknown reward functions.

At a technical level, our work is related to Amin et al. [4] and Abernethy et al. [1], which also relate KWIK learning to bandit learning in a different context unrelated to fairness.

## 2   Preliminaries

We study the *contextual bandit* setting, defined by a domain $\mathcal{X}$, a set of "arms" $[k] := \{1, \ldots, k\}$ and a class $C$ of functions of the form $f : \mathcal{X} \to [0, 1]$. For each arm $j$ there is some function $f_j \in C$, unknown to the learner. In rounds $t = 1, \ldots, T$, an adversary reveals to the algorithm a *context* $x_j^t$ for each arm[2]. An algorithm $\mathcal{A}$ then chooses an arm $i_t$, and observes stochastic reward $r_{i_t}^t$ for the arm it chose. We assume $r_j^t \sim \mathcal{D}_j^t$, $\mathbb{E}[r_j^t] = f_j(x_j^t)$, for some distribution $\mathcal{D}_j^t$ over $[0, 1]$.

Let $\Pi$ be the set of policies mapping contexts to distributions over arms $X^k \to \Delta^k$, and $\pi^*$ the optimal policy which selects a distribution over arms as a function of contexts to maximize the expected reward of those arms. The **pseudo-regret** of an algorithm $\mathcal{A}$ on contexts $x^1, \ldots, x^T$ is defined as follows, where $\pi^t$ represents $\mathcal{A}$'s distribution on arms at round $t$:

$$\sum_t \mathbb{E}_{i_*^t \sim \pi^*(x^t)}[f_{i_*^t}(x_{i_*^t}^t)] - \mathbb{E}_{i^t \sim \pi^t}[\sum_t f_{i^t}(x_{i^t}^t)] = \text{Regret}(x^1, \ldots, x^T),$$

shorthanded as $\mathcal{A}$'s **regret**. Optimal policy $\pi^*$ pulls arms with highest expectation at each round, so:

$$\text{Regret}(x^1, \ldots, x^T) = \sum_t \max_j \left( f_j(x_j^t) \right) - \mathbb{E}_{i^t \sim \pi^t}[\sum_t f_{i^t}(x_{i^t}^t)].$$

We say that $\mathcal{A}$ satisfies regret bound $R(T)$ if $\max_{x^1, \ldots, x^T} \text{Regret}(x^1, \ldots, x^t) \leq R(T)$.

Let the history $h^t \in \left( \mathcal{X}^k \times [k] \times [0, 1] \right)^{t-1}$ be a record of $t - 1$ rounds experienced by $\mathcal{A}$, $t - 1$ 3-tuples encoding the realization of the contexts, arm chosen, and reward observed. $\pi_{j|h^t}^t$ denotes the probability that $\mathcal{A}$ chooses arm $j$ after observing contexts $x^t$, given $h^t$. For simplicity, we will often drop the superscript $t$ on the history when referring to the distribution over arms: $\pi_{j|h}^t := \pi_{j|h^t}^t$.

We now define what it means for a contextual bandit algorithm to be $\delta$-fair with respect to its arms. Informally, this will mean that $\mathcal{A}$ will play arm $i$ with higher probability than arm $j$ in round $t$ only if $i$ has higher mean than $j$ in round $t$, for all $i, j \in [k]$, and in all rounds $t$.

**Definition 1** ($\delta$-fair). $\mathcal{A}$ is $\delta$-**fair** if, for all sequences of contexts $x^1, \ldots, x^t$ and all payoff distributions $\mathcal{D}_1^t, \ldots, \mathcal{D}_k^t$, with probability at least $1 - \delta$ over the realization of the history $h$, for all rounds $t \in [T]$ and all pairs of arms $j, j' \in [k]$,

$$\pi_{j|h}^t > \pi_{j'|h}^t \text{ only if } f_j(x_j^t) > f_{j'}(x_{j'}^t).$$

**KWIK learning**   Let $\mathcal{B}$ be an algorithm which takes as input a sequence of examples $x^1, \ldots, x^T$, and when given some $x^t \in \mathcal{X}$, outputs either a prediction $\hat{y}^t \in [0, 1]$ or else outputs $\hat{y}^t = \bot$, representing "I don't know". When $\hat{y}^t = \bot$, $\mathcal{B}$ receives feedback $y^t$ such that $\mathbb{E}[y^t] = f(x^t)$. $\mathcal{B}$ is an $(\epsilon, \delta)$-KWIK learning algorithm for $C : \mathcal{X} \to [0, 1]$, with KWIK bound $m(\epsilon, \delta)$ if for any sequence of examples $x^1, x^2, \ldots$ and any target $f \in C$, with probability at least $1 - \delta$, both:

1. Its numerical predictions are accurate: for all $t$, $\hat{y}^t \in \{\bot\} \cup [f(x^t) - \epsilon, f(x^t) + \epsilon]$, and
2. $\mathcal{B}$ rarely outputs "I Don't Know": $\sum_{t=1}^{\infty} \mathbb{I}[\hat{y}^t = \bot] \leq m(\epsilon, \delta)$.

## 2.1 Specializing to Classic Stochastic Bandits

In Sections 3.1 and 3.2, we study the classic stochastic bandit problem, an important special case of the contextual bandit setting described above. Here we specialize our notation to this setting, in which there are no contexts. For each arm $j \in [k]$, there is an unknown distribution $\mathcal{D}_j$ over $[0,1]$ with unknown mean $\mu_j$. A learning algorithm $\mathcal{A}$ chooses an arm $i_t$ in round $t$, and observes the reward $r_{i_t}^t \sim \mathcal{D}_{i_t}$ for the arm that it chose. Let $i^* \in [k]$ be the arm with highest expected reward: $i^* \in \arg\max_{i \in [k]} \mu_i$. The pseudo-regret of an algorithm $\mathcal{A}$ on $\mathcal{D}_1, \ldots, \mathcal{D}_k$ is now just:

$$T \cdot \mu_{i^*} - \mathbb{E}_{i^t \sim \pi^t} \Big[ \sum_{0 \le t \le T} \mu_{i^t} \Big] = \text{Regret}(T, \mathcal{D}_1, \ldots, \mathcal{D}_k)$$

Let $h^t \in ([k] \times [0,1])^{t-1}$ denote a record of the $t-1$ rounds experienced by the algorithm so far, represented by $t-1$ 2-tuples encoding the previous arms chosen and rewards observed. We write $\pi_{j|h^t}^t$ to denote the probability that $\mathcal{A}$ chooses arm $j$ given history $h^t$. Again, we will often drop the superscript $t$ on the history when referring to the distribution over arms: $\pi_{j|h}^t := \pi_{j|h^t}^t$. $\delta$-fairness in the classic bandit setting then specializes as follows:

**Definition 2** ($\delta$-fairness in the classic bandits setting). $\mathcal{A}$ is $\delta$-**fair** if, for all distributions $\mathcal{D}_1, \ldots, \mathcal{D}_k$, with probability at least $1 - \delta$ over the history $h$, for all $t \in [T]$ and all $j, j' \in [k]$:

$$\pi_{j|h}^t > \pi_{j'|h}^t \text{ only if } \mu_j > \mu_{j'}.$$

## 3 Classic Bandits Setting

### 3.1 Fair Classic Stochastic Bandits: An Algorithm

In this section, we describe a simple and intuitive modification of the standard UCB algorithm [5], called FAIRBANDITS, prove that it is fair, and analyze its regret bound. The algorithm and its analysis highlight a key idea that is important to the design of fair algorithms in this setting: that of *chaining* confidence intervals. Intuitively, as a $\delta$-fair algorithm explores different arms it must play two arms $j_1$ and $j_2$ with equal probability until it has sufficient data to deduce, with confidence $1 - \delta$, either that $\mu_{j_1} > \mu_{j_2}$ or vice versa. FAIRBANDITS does this by maintaining empirical estimates of the means of both arms, together with confidence intervals around those means. To be safe, the algorithm must play the arms with equal probability while their confidence intervals overlap. The same reasoning applies simultaneously to every pair of arms. Thus, if the confidence intervals of each pair of arms $j_i$ and $j_{i+1}$ overlap for each $i \in [k]$, the algorithm is forced to play *all* arms $j$ with equal probability. This is the case even if the confidence intervals around arm $j_k$ and arm $j_1$ are far from overlapping – i.e. when the algorithm can be confident that $\mu_{j_1} > \mu_{j_k}$.

This chaining approach initially seems overly conservative when ruling out arms, as reflected in its regret bound, which is only non-trivial after $T \gg k^3$. In contrast, the UCB algorithm [5] achieves non-trivial regret after $T = O(k)$ rounds. However, our lower bound in Section 3.2 shows that *any* fair algorithm *must* suffer constant per-round regret for $T \gg k^3$ rounds on some instances.

We now give an overview of the behavior of FAIRBANDITS. At every round $t$, FAIRBANDITS identifies the arm $i_*^t = \arg\max_i u_i^t$ that has the largest *upper* confidence interval amongst the active arms. At each round $t$, we say $i$ is *linked* to $j$ if $[\ell_i^t, u_i^t] \cap [\ell_j^t, u_j^t] \neq \emptyset$, and $i$ is *chained* to $j$ if $i$ and $j$ are in the same component of the transitive closure of the linked relation. FAIRBANDITS plays uniformly at random among all active arms chained to arm $i_*^t$.

Initially, the active set contains all arms. The active set of arms at each subsequent round is defined to be the set of arms that are chained to the arm with highest upper confidence bound at the previous round. The algorithm can be confident that arms that have become unchained to the arm with the highest upper confidence bound at any round have means that are lower than the means of any chained arms, and hence such arms can be safely removed from the active set, never to be played again. This has the useful property that the active set of arms can only shrink: at any round $t$, $S_t \subseteq S_{t-1}$.

We first observe that with probability $1 - \delta$, all of the confidence intervals maintained by FAIRBANDITS ($\delta$) contain the true means of their respective arms over all rounds. We prove this claim, along with all other claims in this paper, in the full technical version of this paper [17].

```
 1: procedure FAIRBANDITS(δ)
 2:     S^0 ← {1, . . . , k}                                              ▷ Initialize the active set
 3:     for i = 1, . . . k do
 4:         μ̂_i^0 ← 1/2, u_i^0 ← 1, ℓ_i^0 ← 0, n_i^0 ← 0                  ▷ Initialize each arm
 5:     for t = 1 to T do
 6:         i_*^t ← arg max_{i∈S^{t-1}} u_i^t                            ▷ Find arm with highest ucb
 7:         S^t ← {j | j chains to i_*^t, j ∈ S^{t-1}}                    ▷ Update active set
 8:         j^* ← (x ∈_R S^t)                                           ▷ Select active arm at random
 9:         n_{j^*}^{t+1} ← n_{j^*}^t + 1
10:         μ̂_{j^*}^{t+1} ← 1/n_{j^*}^{t+1} (μ̂_{j^*}^t · n_{j^*}^t + r_{j^*}^t)   ▷ Pull arm j^*, update its mean estimate
11:         B ← √( ln((π·(t+1))^2/3δ) / 2n_{j^*}^{t+1} )
12:         [ℓ_{j^*}^{t+1}, u_{j^*}^{t+1}] ← [μ̂_{j^*}^{t+1} − B, μ̂_{j^*}^{t+1} + B]   ▷ Update interval for pulled arm
13:         for j ∈ S^t, j ≠ j^* do
14:             μ̂_j^{t+1} ← μ̂_j^t, n_j^{t+1} ← n_j^t, u_j^{t+1} ← u_j^t, ℓ_j^{t+1} ← ℓ_j^t
```

**Lemma 1.** *With probability at least $1 − δ$, for every arm $i$ and round $t$ $\ell_i^t \leq \mu_i \leq u_i^t$.*

The fairness of FAIRBANDITS follows: with high probability the algorithm constructs good confidence intervals, so it can confidently choose between arms without violating fairness.

**Theorem 1.** FAIRBANDITS $(δ)$ *is $δ$-fair.*

Having proven that FAIRBANDITS is indeed fair, it remains to upper-bound its regret. We proceed by a series of lemmas, first lower bounding the probability that any arm active in round $t$ has been pulled substantially fewer times than its expectation.

**Lemma 2.** *With probability at least $1 − \frac{δ}{2t^2}$, $n_i^t \geq \frac{t}{k} − \sqrt{\frac{t}{2} \ln\left(\frac{2k·t^2}{δ}\right)}$ for all $i \in S^t$ (for all active arms in round $t$).*

We now use this lower bound on the number of pulls of active arm $i$ in round $t$ to upper-bound $\eta(t)$, an upper bound on the confidence interval width FAIRBANDITS uses for any active arm $i$ in round $t$.

**Lemma 3.** *Consider any round $t$ and any arm $i \in S^t$. Condition on $n_i^t \geq \frac{t}{k} − \sqrt{\frac{t \ln(\frac{2kt^2}{δ})}{2}}$. Then,*

$$u_i^t − \ell_i^t \leq 2\sqrt{\frac{\ln\left((\pi · t)^2/3δ\right)}{2 · \frac{t}{k} − \sqrt{\frac{t \ln(\frac{2kt^2}{δ})}{2}}}} = \eta(t).$$

We stitch together these lemmas as follows: Lemma 2 upper bounds the probability that any arm $i$ active in round $t$ has been pulled substantially fewer times than its expectation, and Lemma 3 upper bounds the width of any confidence interval used by FAIRBANDITS in round $t$ by $\eta(t)$. Together, these enable us to determine how both the number of arms in the active set, as well as the spread of their confidence intervals, evolve over time. This translates into the following regret bound.

**Theorem 2.** *If $δ < 1/\sqrt{T}$, then* FAIRBANDITS *has regret $R(T) = O\left(\sqrt{k^3 T \ln \frac{Tk}{δ}}\right)$.*

Two points are worth highlighting in Theorem 2. First, this bound becomes non-trivial (i.e. the average per-round regret is $\ll 1$) for $T = \Omega(k^3)$. As we show in the next section, it is not possible to improve on this. Second, the bound may appear to have suboptimal dependence on $T$ when compared to unconstrained regret bounds (where the dependence on $T$ is often described as logarithmic). However, it is known that $\Omega\left(\sqrt{kT}\right)$ regret is *necessary* even in the unrestricted setting (without fairness) if one does not make data-specific assumptions on an instance [9] It would be possible to state a logarithmic dependence on $T$ in our setting as well while making assumptions on the gaps between arms, but since our fairness constraint manifests itself as a cost that depends on $k$, we choose for clarity to avoid such assumptions; without them, our dependence on $T$ is also optimal.

## 3.2 Fair Classic Stochastic Bandits: A Lower Bound

We now show that the regret bound for FAIRBANDITS has an optimal dependence on $k$: *no* fair algorithm has diminishing regret before $T = \Omega(k^3)$ rounds. At a high level, we construct our lower bound example to embody the "worst of both worlds" for fair algorithms: the arm payoff means are just close enough together that the chain takes a long time to break, and the arm payoff means are just far enough apart that the algorithm incurs high regret while the chain remains unbroken. This lets us prove the formal statement below. The full proof, which proceeds via Bayesian reasoning using priors for the arm means, may be found in our technical companion paper [17].

**Theorem 3.** *There is a distribution $P$ over $k$-arm instances of the stochastic multi-armed bandit problem such that any fair algorithm run on $P$ experiences constant per-round regret for at least $T = \Omega\left(k^3 \ln \frac{1}{\delta}\right)$ rounds.*

Thus, we tightly characterize the optimal regret attainable by fair algorithms in the classic bandits setting, and formally separate it from the regret attainable by algorithms absent a fairness constraint. Note that this already shows a separation between the best possible learning rates for contextual bandit learning with and without the fairness constraint – the classic multi-armed bandit problem is a special case of every contextual bandit problem, and for general contextual bandit problems, it is also known how to get non-trivial regret after only $T = O(k)$ many rounds [3, 8, 12].

# 4 Contextual Bandits Setting

## 4.1 KWIK Learnability Implies Fair Bandit Learnability

In this section, we show if a class of functions is KWIK learnable, then there is a fair algorithm for learning the same class of functions in the contextual bandit setting, with a regret bound polynomially related to the function class' KWIK bound. Intuitively, KWIK-learnability of a class of functions guarantees we can learn the function's behavior to a high degree of accuracy with a high degree of confidence. As fairness constrains an algorithm most before the algorithm has determined the payoff functions' behavior accurately, this guarantee enables us to learn fairly without incurring much additional regret. Formally, we prove the following polynomial relationship.

**Theorem 4.** *For an instance of the contextual multi-armed bandit problem where $f_j \in C$ for all $j \in [k]$, if $C$ is $(\epsilon, \delta)$-KWIK learnable with bound $m(\epsilon, \delta)$, KWIKToFAIR $(\delta, T)$ is $\delta$-fair and achieves regret bound:*

$$R(T) = O\left(\max\left(k^2 \cdot m\left(\epsilon^*, \frac{\min(\delta, 1/T)}{T^2 k}\right), k^3 \ln \frac{k}{\delta}\right)\right)$$

*for $\delta \leq \frac{1}{\sqrt{T}}$ where $\epsilon^* = \arg\min_\epsilon(\max(\epsilon \cdot T, k \cdot m(\epsilon, \frac{\min(\delta, 1/T)}{kT^2})))$.*

First, we construct an algorithm KWIKToFAIR$(\delta, T)$ that uses the KWIK learning algorithm as a subroutine, and prove that it is $\delta$-fair. A call to KWIKToFAIR$(\delta, T)$ will initialize a KWIK learner for each arm, and in each of the $T$ rounds will implicitly construct a confidence interval around the prediction of each learner. If a learner makes a numeric valued prediction, we will interpret this as a confidence interval centered at the prediction with width $\epsilon^*$. If a learner outputs $\perp$, we interpret this as a trivial confidence interval (covering all of $[0, 1]$). We then use the same chaining technique used in the classic setting to choose an arm from the set of arms chained to the predicted top arm. Whenever all learners output predictions, they need no feedback. When a learner for $j$ outputs $\perp$, *if $j$ is selected* then we have feedback $r_j^t$ to give it; on the other hand, if $j$ isn't selected, we "roll back" the learning algorithm for $j$ to before this round by not updating the algorithm's state.

```
1: procedure KWIKToFAIR(δ, T)
2:     δ* ← min(δ, 1/T)/(kT²), ε* ← arg min_ε(max(ε · T, k · m(ε, δ*)))
3:     Initialize KWIK(ε*, δ*)-learner L_i, h_i ← [ ] ∀i ∈ [k]
4:     for 1 ≤ t ≤ T do
5:         S ← ∅                                              ▷ Initialize set of predictions S
6:         for i = 1, . . . , k do
7:             s_i^t ← L_i(x_i^t, h_i)
8:             S ← S ∪ s_i^t                                  ▷ Store prediction s_i^t
```

```
9:        if ⊥ ∈ S then
10:            Pull j* ← (x ∈_R [k]), receive reward r^t_{j*}          ▷ Pick arm at random from all arms
11:        else
12:            i^t_* ← arg max_i s^t_i
13:            S^t ← {j | (s^t_j − ε*, s^t_j + ε*) chains to (s^t_{i^t_*} − ε*, s^t_{i^t_*} + ε*)}
14:            Pull j* ← (x ∈_R S^t), receive reward r^t_{j*}   ▷ Pick arm at random from active set s^t_{i^t_*}
15:        h_{j*} ← h_{j*} :: (x^t_{j*}, r^t_{j*})                      ▷ Update the history for L_{j*}
```

We begin by bounding the probability of certain failures of KWIKTOFAIR in Lemma 4.

**Lemma 4.** *With probability at least* $1 - \min(\delta, \frac{1}{T})$, *for all rounds $t$ and all arms $i$, (a) if $s^t_i \in \mathbb{R}$ then* $|s^t_i - f_i(x^t_i)| \le \epsilon^*$ *and (b)* $\sum_t \mathbb{I}[s^t_i = \perp \text{ and } i \text{ is pulled}] \le m(\epsilon^*, \delta^*)$.

This in turn lets us prove the fairness of KWIKTOFAIR in Theorem 5. Intuitively, the KWIK algorithm's confidence about predictions translates into confidence about expected rewards, which lets us choose between arms without violating fairness.

**Theorem 5.** KWIKTOFAIR$(\delta, T)$ *is $\delta$-fair.*

We now use the KWIK bounds of the KWIK learners to upper-bound the regret of KWIKTO-FAIR$(\delta, T)$. We proceed by bounding the regret incurred in those rounds when all KWIK algorithms make a prediction (i.e., when we have a nontrivial confidence interval for each arm's expected reward) and then bounding the number of rounds for which some learner outputs $\perp$ (i.e., when we choose randomly from all arms and thus incur constant regret). These results combine to produce Lemma 5.

**Lemma 5.** KWIKTOFAIR$(\delta, T)$ *achieves regret* $O(\max(k^2 \cdot m(\epsilon^*, \delta^*), k^3 \ln \frac{Tk}{\delta}))$.

Our presentation of KWIKTOFAIR$(\delta, T)$ has a known time horizon $T$. Its guarantees extend to the case in which $T$ is unknown via the standard "doubling trick" to prove Theorem 4.

An important instance of the contextual bandit problem is the linear case, where $C$ consists of the set of all *linear* functions of bounded norm in $d$ dimensions, i.e. when the rewards of each arm are governed by an underlying linear regression model on contexts. Known KWIK algorithms [26] for the set of linear functions $C$ then allow us, via our reduction, to give a fair contextual bandit algorithm for this setting with a polynomial regret bound.

**Lemma 6** ([26]). *Let* $C = \{f_\theta | f_\theta(x) = \langle \theta, x \rangle, \theta \in \mathbb{R}^d, ||\theta|| \le 1\}$ *and* $\mathcal{X} = \{x \in \mathbb{R}^d : ||x|| \le 1\}$. *$C$ is KWIK learnable with KWIK bound* $m(\epsilon, \delta) = \tilde{O}(d^3/\epsilon^4)$.

Then, an application of Theorem 4 implies that KWIKTOFAIR has a polynomial regret guarantee for the class of linear functions.

**Corollary 1.** *Let $C$ and $\mathcal{X}$ be as in Lemma 6, and $f_j \in C$ for each $j \in [k]$. Then,* KWIKTO-FAIR$(T, \delta)$ *using the learner from [26] has regret* $R(T) = \tilde{O}\left(\max\left(T^{4/5}k^{6/5}d^{3/5}, k^3 \ln \frac{k}{\delta}\right)\right)$.

## 4.2  Fair Bandit Learnability Implies KWIK Learnability

In this section, we show how to use a fair, no-regret contextual bandit algorithm to construct a KWIK learning algorithm whose KWIK bound has logarithmic dependence on the number of rounds $T$. Intuitively, any fair algorithm which achieves low regret must both be able to find and exploit an optimal arm (since the algorithm is no-regret) *and* can only exploit that arm once it has a tight understanding of the qualities of all arms (since the algorithm is fair). Thus, any fair no-regret algorithm will ultimately have tight $(1 - \delta)$-confidence about each arm's reward function.

**Theorem 6.** *Suppose $\mathcal{A}$ is a $\delta$-fair algorithm for the contextual bandit problem over the class of functions $C$, with regret bound $R(T, \delta)$. Suppose also there exists $f \in C, x(\ell) \in \mathcal{X}$ such that for every $\ell \in [\lceil \frac{1}{\epsilon} \rceil]$, $f(x(\ell)) = \ell \cdot \epsilon$. Then,* FAIRTOKWIK *is an $(\epsilon, \delta)$-KWIK algorithm for $C$ with KWIK bound $m(\epsilon, \delta)$, with $m(\epsilon, \delta)$ the solution to* $\frac{m(\epsilon, \delta)\epsilon}{4} = R(m(\epsilon, \delta), \frac{\epsilon\delta}{2T})$.

*Remark* 1. The condition that $C$ should contain a function that can take on values that are multiples of $\epsilon$ is for technical convenience; $C$ can always be augmented by adding a single such function.

Our aim is to construct a KWIK algorithm $\mathcal{B}$ to predict labels for a sequence of examples labeled with some unknown function $f^* \in C$. We provide a sketch of the algorithm, FAIRTOKWIK, below, and refer interested readers to our full technical paper [17] for a complete and formal description.

We use our fair algorithm to construct a KWIK algorithm as follows: we will run our fair contextual bandit algorithm $\mathcal{A}$ on an instance that we construct online as examples $x^t$ arrive for $\mathcal{B}$. The idea is to simulate a two arm instance, in which one arm's rewards are governed by $f^*$ (the function to be KWIK learned), and the other arm's rewards are governed by a function $f$ that we can set to take any value in $\{0, \epsilon, 2\epsilon, \ldots, 1\}$. For each input $x^t$, we perform a thought experiment and consider $\mathcal{A}$'s probability distribution over arms when facing a context which forces arm 2's payoff to take each of the values $0, \epsilon^*, 2\epsilon^*, \ldots, 1$. Since $\mathcal{A}$ is fair, $\mathcal{A}$ will play arm 1 with weakly higher probability than arm 2 for those $\ell : \ell\epsilon^* \leq f(x^t)$; analogously, $\mathcal{A}$ will play arm 1 with weakly lower probability than arm 2 for those $\ell : \ell\epsilon^* \geq f(x^t)$. If there are at least 2 values of $\ell$ for which arm 1 and arm 2 are played with equal probability, one of those contexts will force $\mathcal{A}$ to suffer $\epsilon^*$ regret, so we continue the simulation of $\mathcal{A}$ on one of those instances selected at random, forcing at least $\epsilon^*/2$ regret in expectation, and at the same time have $\mathcal{B}$ return $\perp$. $\mathcal{B}$ receives $f^*(x^t)$ on such a round, which is used to construct feedback for $\mathcal{A}$. Otherwise, $\mathcal{A}$ must transition from playing arm 1 with strictly higher probability to playing 2 with strictly higher probability as $\ell$ increases: the point at which that occurs will "sandwich" the value of $f(x^t)$, since $\mathcal{A}$'s fairness implies this transition must occur when the expected payoff of arm 2 exceeds that of arm 1. $\mathcal{B}$ uses this value to output a numeric prediction.

An important fact we exploit is that we can *query* $\mathcal{A}$'s behavior on $(x^t, x(\ell))$, for any $x^t$ and $\ell \in \left[\left\lceil \frac{1}{\epsilon^*} \right\rceil\right]$ without providing it feedback (and instead "roll back" its history to $h^t$ not including the query $(x^t, x(\ell))$). We update $\mathcal{A}$'s history by providing it feedback only in rounds where $\mathcal{B}$ outputs $\perp$. Finally, we note that, as in KWIKTOFAIR, the claims of FAIRTOKWIK extend to the infinite horizon case via the doubling trick.

### 4.3 An Exponential Separation Between Fair and Unfair Learning

In this section, we use this Fair-KWIK equivalence to give a simple contextual bandit problem for which fairness imposes an *exponential* cost in its regret bound, unlike the polynomial cost proven in the linear case in Section 4.1. In this problem, the context domain is the $d$-dimensional boolean hypercube: $\mathcal{X} = \{0, 1\}^d$ – i.e. the context each round for each individual consists of $d$ boolean attributes. Our class of functions $C$ is the class of boolean *conjunctions*: $C = \{f \mid f(x) = x_{i_1} \wedge x_{i_2} \wedge \ldots \wedge x_{i_k}$ where $0 \leq k \leq d$ and $i_1, \ldots, i_k \in [d]\}$.

We first note that there exists a simple but unfair algorithm for this problem which obtains regret $R(T) = O(k^2 d)$. A full description of this algorithm, called CONJUNCTIONBANDIT, may be found in our technical companion paper [17]. We now show that, in contrast, fair algorithms cannot guarantee subexponential regret in $d$. This relies upon a known lower bound for KWIK learning conjunctions [21]:

**Lemma 7.** *There exists a sequence of examples $(x^1, \ldots, x^{2^d-1})$ such that for $\epsilon, \delta \leq 1/2$, every $(\epsilon, \delta)$-KWIK learning algorithm $\mathcal{B}$ for the class $C$ of conjunctions on $d$ variables must output $\perp$ for $x^t$ for each $t \in [2^d - 1]$. Thus, $\mathcal{B}$ has a KWIK bound of at least $m(\epsilon, \delta) = \Omega(2^d)$.*

We then use the equivalence between fair algorithms and KWIK learning to translate this lower bound on $m(\epsilon, \delta)$ into a minimum worst case regret bound for fair algorithms on conjunctions. We modify Theorem 6 to yield the following lemma.

**Lemma 8.** *Suppose $\mathcal{A}$ is a $\delta$-fair algorithm for the contextual bandit problem over the class $C$ of conjunctions on $d$ variables. If $\mathcal{A}$ has regret bound $R(T, \delta)$ then for $\delta' = 2T\delta$, FAIRTOKWIK is an $(0, \delta')$-KWIK algorithm for $C$ with KWIK bound $m(0, \delta') = 4R(m(0, \delta'), \delta)$.*

Lemma 7 then lets us lower-bound the worst case regret of fair learning algorithms on conjunctions.

**Corollary 2.** *For $\delta < \frac{1}{2T}$, any $\delta$-fair algorithm for the contextual bandit problem over the class $C$ of conjunctions on $d$ boolean variables has a worst case regret bound of $R(T) = \Omega(2^d)$.*

Together with the analysis of CONJUNCTIONBANDIT, this demonstrates a strong separation between fair and unfair contextual bandit algorithms: when the underlying functions mapping contexts to payoffs are conjunctions on $d$ variables, there exist a sequence of contexts on which fair algorithms must incur regret exponential in $d$ while unfair algorithms can achieve regret linear in $d$.

## Footnotes

[2]Often, the contextual bandit problem is defined such that there is a single context $x^t$ every day. Our model is equivalent – we could take $x_j^t := x^t$ for each $j$.

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
