[Reviews · NeurIPS 2016]

Reviewer 1

Summary

This paper continues the work of the first first paper on giving a new definition of fairness in learning, extending the setting to contextual bandits. This is, of course, the first natural step in making fair algorithms more interesting and practical. The first result in this paper is that the existence of a "KWIK" algorithm implies the existence of a fair contextual bandit algorithm for the same class of functions. The proof is, of course, a reduction, and it uses similar ideas to the proof in Part I. (To think of it, it also reminds me of RandomizedUCB from Dudik et al). A similar regret bound is proven here. More surprising, to me, is the reduction in the other direction -- a fair algorithm implies a KWIK one, again via a reduction. Very roughly speaking, the simulation causes the fair algorithm to give accurate estimates of its own performance by ``sandwiching" the estimate within an epsilon range of the true value. Finally an exponential separation is proven between fair and regular bandit learning in the case of conjunctions. This class of functions does not appear so practically interesting, but is rather a "learning theory" result on the KWIK framework. This is exactly the type of work I expect part I (the first paper of the two) to inspire. Much of the novelty of this paper comes from the first part, which was submitted separately. However I do like the connection given here to the KWIK framework. I think that because so much of this paper is spent re-justifying the model, and because the reduction in one direction feel similar, it would not be so crazy to join both parts into one paper. However, I think this can also stand alone as a separate contribution, though in that case, it is closer to the accept/reject boundary.

Qualitative Assessment

A nice extension to the contextual bandit problem and relationship to KWIK. If not for high overlap with Part I, the ratings would be higher.

Confidence in this Review

2-Confident (read it all; understood it all reasonably well)


Reviewer 2

Summary

As in the paper devoted to classic bandits, the article develops the notion of fairness but in the setting of contextual bandits this time. Equivalence between KWIK and fair contextual bandit learnabilities is proved. Introducing KWIK learnability is an elegant trick to solve fair contextual bandits.

Qualitative Assessment

In section 4.1, an exponential separation is proved between fair and unfair learning in terms of the dimension d of the context set. What are the separations in terms of the horizon T and the number of arms k? Beyond this point, again style should be improved, the paper is too dense to be readable.

Confidence in this Review

3-Expert (read the paper in detail, know the area, quite certain of my opinion)


Reviewer 3

Summary

The paper considers contextual bandit algorithms with a fairness guarantee. The rewards are assumed to be stochastic and the guarantee demanded is that the probability of choosing arm A over B is never higher if the expected payoff of A is greater than that of B (whp). This is considered in a contextual bandit setting (as opposed to the standard one), where for each arm a context $x^t_a$ is provided at time $t$. And the expected payoff is some f(x^t_a) for some function f coming from some class.

Qualitative Assessment

The paper compares this 'fair' contextual bandit problem to the so called KWIK-learning framework. In the KWIK framework, the online learning algorithm, either outputs an answer, or outputs 'I don't know'. It only receives feedback (a r.v. with expectation f(x_t)) if it outputs 'I don't know'. The performance requirement is that all predictions are $\epsilon$-accurate and the number of times the algorithm outputs 'I don't know' is bounded. The paper shows that KWIK-learnability implies fair contextual bandit. This result is nice even if the proof is along the lines that one might expect. They also show a converse that a fair contextual-bandit algorithm with good regret implies KWIK learnability. - After reading the author feedback, I'm convinced of the correctness of the other result. Based on that I recommend acceptance.

Confidence in this Review

2-Confident (read it all; understood it all reasonably well)


Reviewer 4

Summary

This paper extends their study on fair learning in another submission (paper #212) to contextual bandit setting. The main contribution of this submission is revealing the dependency on the context dimension $d$ for the regret of a fair learning algorithm. The main technical tool is the KWIK learning model. By establishing an equivalent relationship between fairness and the KWIK learning model, the authors show that the regret of a fair learning algorithm for linear contextual bandits is polynomial on $d$, while it is exponential for ConjunctionBandit.

Qualitative Assessment

Following another submission (paper #212), this paper studies the fair learning algorithms in contextual bandits. Compared to paper #212, this submission investigates the dependency of the regret on the context dimension, by connecting the fairness to the KWIK learning model. The paper is well written in most parts: it clearly motivates the problem, discusses the relevant previous work, and states the main contributions. Some parts in the preliminaries should be clarified. Some more intuitions about the KWIK learning model and the theoretical results can be discussed in more details: - Line 129 claims to focus on the “stochastic contextual bandit setting” while Line 131 says “an adversary reveals … ”. Do you actual study stochastic or adversarial contextual bandits? - Lines 155 – 161: after discussing about the KWIK learning model, some intuitive discussions about the KWIK learnability may be needed. For example, what will the bound m(\epsilon, \delta) depend on? A discussion on m(\epsilon, \delta) for the linear case will make it more natural to obtain Lemma 3. - The FairToKWIK algorithm also needs to know the horizon T. Do we need the doubling trick for infinite horizon setting? - Do Lemma 4 and Corollary 2 hold for stochastic or adversarial contextual bandits? Are they true if we consider i.i.d. arrival contexts? After the Rebuttal and Discussion: Given the differences in the contents and techniques from #212, treating this part as a separate contribution is acceptable. To make this submission less overlapped with #212, this part needs to highlight the differences in the introduction section: 1) motivate the introduction of contexts; 2) discuss the challenges in contextual setting; 3) main contributions in this setting (this part is already done in the original submission).

Confidence in this Review

2-Confident (read it all; understood it all reasonably well)


Reviewer 5

Summary

This paper introduces a notion of fairness in contextual bandits. For any pair of arms, w.p. >= 1-delta, a delta-fair algorithm cannot choose an arm with a lower expected reward. The main result of this paper is that it establishes the connection between fairness and the KWIK learning framework. First, it is shown that given a KWIK learner, it is possible to construct a fair algorithm whose regret bound depends on the KWIK bound. Conversely, given a fair algorithm a KWIK learning algorithm is proposed with a particular dependence on the regret bound of the fair algorithm. An intuitive explanation of these connections is that a delta-fair algorithm necessarily has to estimate the true reward function to delta-accuracy, from which the KWIK learnability also follows. Using this so called equivalence, an exponential lower bound for the regret in the fair case is shown for boolean contexts. That is, the cost of fairness may be prohibitively large for contextual bandits, as opposed to the stochastic case where we only suffered O(k^2).

Qualitative Assessment

- There is a significant overlap between the two submissions on fairness. Given the level of mathematical detail presented in the main text, it is my view that the two manuscripts can be amalgamated into a single submission, even given the 8 page limit. - The authors have done a good job of presenting the mathematical intuition underlying the connections. - In the statement of Theorem 3, it would help to have a concrete example where one can explicitly write down the equation connecting the KWIK bound and the regret bound in the fair case. The linear case would be good. - Line 256: use \citet. Line 271: use \{ 1, … , \left[ \ceil{…} \right] \} instead of \left[ \ceil{…} \right] which is a bit awkward.

Confidence in this Review

2-Confident (read it all; understood it all reasonably well)


Reviewer 6

Summary

This paper also studied the fairness in learning algorithm, along with another submission paper ID 212 which I also reviewed. Paper ID 212 studied the case with classical multi-armed bandit. This paper considered the more general contextual bandit case, where a contextual bandit algorithm is considered \delta-fair, if it never selects an option with probability strictly greater than another better option. The paper establish an equivalence between fair contextual multi-armed bandit and the KWIK learning. Roughly speaking, if a function class C is KWIK learnable, then the corresponding fair contextual multi-armed bandit problem can achieves a non-trivial regret after a polynomial number of rounds; and if the fair contextual multi-armed bandit problem admits a non-trivial algorithm, then the function class C is KWIK learnable. The proved guarantee is rather complicated, and indeed establish that these two concept are polynomially related. In order to show that, the paper first gives a simple algorithm for the fair contextual multi-armed bandit problem. The idea is very simple, it associates a learner with carefully chosen parameters for each arm, and then it use the learners to provide a confidence range for each arm. In order to be fair, the algorithm treat all arms with overlapping confidence range as equal, and thus crate an equivalence class between the arms. Then it pick a random arm within the equivalent class containing the arm with highest upper range, and feed a sample to that. There are some technical points like rolling back the history for the learner in order to deal with the bandit feedback setting, but overall the algorithm is intuitive and elegant. Then the paper gives an algorithm for KWIT using a fair contextual multi-armed bandit problem as a subroutine. The high-level idea is that intuitively, since the algorithm is fair, it can only exploit if it is quite certain about the hidden function f. The reduction is also straightforward and the analysis is elegant.

Qualitative Assessment

Technical quality: I am confident that the results are correct. But there are some issues in the proof. The (b) part in Lemma 1 seems not right, it should be the total number of times that s_i^t is not I don't know and i does get sample instead of the stated one, which seems wrong to me. Also, in Lemma 1 it says arm j but it use i later. But the above issue does not affect the validity of the later proof, as it uses the right version. But this somehow makes me misunderstand something when I first read that. Novelty/originality: The study of the fairness in learning seems both interesting and of practical importance to me. And their definition is clearly original. The connection between fair contextual multi-armed bandit and KWIK learning is a fantastic result. It is also nice to know that there is an exponential separation between fair and non-fair algorithm when considering the contextual problems. Potential impact or usefulness: I expect that this paper along with the author's another paper 212 will influence a number of future works in addressing the fairness concern in learning: i.e., maybe their definition can be applied to settings other than contextual multi-armed bandit, or it can inspired some other definition of fairness in learning, which has a weaker impact on the regret achievable. Clarity and presentation: The paper is generally well-written, but there are many typos/errors in some part of the paper. A partial list of typos/errors: As said before, the statement for Lemma 1 is not correct and need to be fix. In line 15 of the KWIKTOFAIR algorithm, it should be "update the history for L_{j*}" One Line 230, the total regret should be 2k\eps^* \cdot T + n +\delta T. On the first line on page 6, n_i seems to mean the number of rounds that arm i outputs I don't know, instead of "some arm". There is a unnecessary ) in the equation after line 237. On line 9 of the FAIRTOKWIK algorithm, the x in A(h^t,x^t,x) should be x(\hat{\elk}). I feel like the submission still needs some revisions.

Confidence in this Review

2-Confident (read it all; understood it all reasonably well)